# Uptake of Cerium Dioxide Nanoparticles and Impact on Viability, Differentiation and Functions of Primary Trophoblast Cells from Human Placenta

**DOI:** 10.3390/nano10071309

**Published:** 2020-07-03

**Authors:** Margaux Nedder, Sonja Boland, Stéphanie Devineau, Amal Zerrad-Saadi, Jasmina Rogozarski, René Lai-Kuen, Ibtissem Baya, Jean Guibourdenche, Francoise Vibert, Audrey Chissey, Sophie Gil, Xavier Coumoul, Thierry Fournier, Ioana Ferecatu

**Affiliations:** 1Faculté de Pharmacie de Paris, Université de Paris, INSERM UMR-S 1139, 3PHM, F-75006 Paris, France; margaux.nedder@parisdescartes.fr (M.N.); amal.zerrad-saadi@parisdescartes.fr (A.Z.-S.); jasmina.rogozarski@parisdescartes.fr (J.R.); ibtissembaya01@gmail.com (I.B.); jean.guibourdenche@cch.aphp.fr (J.G.); francoise.vibert@parisdescartes.fr (F.V.); audrey.chissey@parisdescartes.fr (A.C.); sophie.gil@parisdescartes.fr (S.G.); thierry.fournier@parisdescartes.fr (T.F.); 2BFA, Université de Paris, UMR 8251, CNRS, F-75013 Paris, France; boland@univ-paris-diderot.fr (S.B.); stephanie.devineau@u-paris.fr (S.D.); 3Faculté de Pharmacie de Paris, Université de Paris, INSERM UMS 025—CNRS UMS 3612, F-75006 Paris, France; rene.lai-kuen@parisdescartes.fr; 4Assistance Publique—Hôpitaux de Paris, Hôpital Cochin, Service d’hormonologie, F-75014 Paris, France; 5Université de Paris, INSERM UMR-S 1124, F-75006 Paris, France; xavier.coumoul@parisdescartes.fr

**Keywords:** nanoparticles, cerium dioxide, human placenta, cytotrophoblasts, syncytiotrophoblasts, cytotoxicity, pregnancy hormones

## Abstract

The human placenta is at the interface between maternal and fetal circulations, and is crucial for fetal development. The nanoparticles of cerium dioxide (CeO_2_ NPs) from air pollution are an unevaluated risk during pregnancy. Assessing the consequences of placenta exposure to CeO_2_ NPs could contribute to a better understanding of NPs’ effect on the development and functions of the placenta and pregnancy outcome. We used primary villous cytotrophoblasts purified from term human placenta, with a wide range of CeO_2_ NPs concentrations (0.1–101 μg/cm^2^) and exposure time (24–72 h), to assess trophoblast uptake, toxicity and impact on trophoblast differentiation and endocrine function. We have shown the capacity of both cytotrophoblasts and syncytiotrophoblasts to internalize CeO_2_ NPs. CeO_2_ NPs affected trophoblast metabolic activity in a dose and time dependency, induced caspase activation and a LDH release in the absence of oxidative stress. CeO_2_ NPs decreased the fusion capacity of cytotrophoblasts to form a syncytiotrophoblast and disturbed secretion of the pregnancy hormones hCG, hPL, PlGF, P4 and E2, in accordance with NPs concentration. This is the first study on the impact of CeO_2_ NPs using human primary trophoblasts that decrypts their toxicity and impact on placental formation and functions.

## 1. Introduction

Over the past decades, the development and utilization of engineered nanomaterials have grown exponentially and various nanoparticles (NPs) are used in an increasing number of consumer products [1,2,3,4,5]. Due to their size being less than 100 nm (ISO/TS 80004:2015), NPs have a large surface-to-volume ratio and confer upon the materials they compose unique physicochemical characteristics, including higher surface reactivity, increased catalytic efficiency, and higher resistance when compared to their bulk material counterparts. Depending on their chemical composition, NPs can have different shapes, agglomeration/aggregation states, surface charges, coating and, for some of them, also distinct oxidation states. Thus, the toxicity may vary greatly between each kind of NP.

Lately, novel NPs of cerium dioxide (CeO_2_ NPs or nanoceria) have attracted the attention of the scientific community because they have been used in a range of industrial, commercial and also biomedical applications [6]. These include applications as fuel additive-combustion catalyst in gasoline, as additives in cigarettes, in self-cleaning ovens, as polishing agents and in UV protection [7]. In addition to these applications, CeO_2_ NPs have also been proposed as oxygen sensors (free radical scavengers) and as potential pharmaceuticals (as both anti- and pro-oxidants), due to the peculiar dual oxidative state of cerium (Ce^3+^ and Ce^4+^) [8,9,10]. However, CeO_2_ NPs are also byproducts of combustion processes and thus are emitted unintentionally in the air. Human exposure to CeO_2_ NPs is mainly by inhalation from ambient pollution, especially because of the recent introduction of diesel fuel-additive nanoceria to increase the efficiency of soot combustion of diesel particulate filters [11]. For example, the concentration of cerium is around 9 μg/L in fuel (18 mg/L Ce in fuel-additive) [12].

The adverse human health effects of exposure to CeO_2_ NPs are mostly unknown because there are few epidemiologic studies available so far. Some authors detected cerium in maternal blood samples (8–70 ng/L) [13,14] and in breast milk (5–65 ng/L) [14] in cohorts from Europe, with the highest concentrations of cerium in the maternal blood and milk recorded in the most polluted cities. Two recent articles with cohorts of 400 and 7367 pregnant women, respectively, reported the presence of the element cerium in maternal serum during pregnancy, which was associated with a 4.73-fold increase risk for fetal neural tube defects [15], and in maternal urine, which was associated with a decrease in neonatal TSH levels [16]. Besides, the Organization of Economic Cooperation and Development (OECD) has included CeO_2_ NPs in the top priority list of the nanomaterials requiring urgent evaluation (OECD 2010). All the related in vitro studies are mostly focused on the evaluation of toxicity in cellular models of the lung, liver or immune system, and today the effects of CeO_2_ NPs during the critical window of pregnancy exposure are mostly unexplored. There are very few data on the placental toxicity of CeO_2_ NPs and they are mostly from animal models during gestation [17] exploring the maternal-versus-fetal distribution and the fetotoxicity [18]. However, to our knowledge, there are no available data about the impact of CeO_2_ NPs from in vitro studies employing human models (e.g., placental models such as cotyledon perfusion, trophoblasts cell lines or primary cells). Nevertheless, data were recently published on other types of NP, including metal oxides, such as TiO_2_ NPs [19,20], gold NPs [21,22,23] and silver NPs [24,25], and they all employed animals, the cotyledon perfusion model (to follow their passage from the maternal to fetal circulation) or trophoblast cell lines (to reveal their toxicity). However, again there is no study of the risk NPs pose to primary human trophoblasts.

The placenta is a transitory organ at the interface between the fetus and the mother and is essential for the adaptation and maintenance of pregnancy, fetal development and growth, especially by its intense endocrine function [26]. The placenta also represents a selective barrier that allows gas and nutrient exchanges between the fetus and the mother. The placenta is composed of consecutive arrangements of villous epithelial and stromal tissues in which fetal vessels are embedded [27]. The chorionic villus is covered by a trophoblast bilayer formed by the outer multinucleated syncytium layer termed the syncytiotrophoblast (ST), which is the functional and structural unit of the placenta, and by the underlying mononuclear villous cytotrophoblast (VCT), which differentiate and fuse together to constantly renew the ST. The ST is the seat of placental functions and represents the endocrine tissue of the placenta, secreting a panoply of hormones such as steroids (progesterone (P4) and estrogens like estradiol (E2)), glycoproteins such as human chorionic gonadotropin (hCG) and human placental lactogen (hPL), and factors like soluble fms-like tyrosine kinase 1 (sFlt1) and placental growth factor (PlGF), which is involved in angiogenesis. However, the placenta also ensures the protection of the fetus against a wide variety of xenobiotics, including drugs and pollutants that may pass through other tissue barriers and reach the placenta through the maternal blood flow. Indeed, the ST is in direct contact with maternal blood and circulating pollutants. Due to their small size, NPs have greater potential to travel through the organism than larger particles [1]. Unlike other chemicals, NPs cannot be metabolized and eliminated by the classic cellular detoxification system and thus they could accumulate within most tissues, such as liver or, in the case of pregnancy, the placenta, certainly reaching higher concentrations than initially in the blood due to tissue retention [17,28]. NPs that accumulate within the placenta may cause toxicity and/or dysfunction of the placental barrier, which ultimately impacts the harmonic development or growth of the fetus, leading to pregnancy complications.

The toxicity of CeO_2_ NPs remains controversial, because conflicting results have been reported in the scientific community, with both pro-oxidant and anti-oxidant effects in tissues. This variability depends mostly on the cell type, but also on the physical and chemical characteristics of nanoceria, like the aggregation state, the shape (nanocubes vs. nanorods), surface modifications and the oxidative status of the cerium (Ce) [29,30]. Altogether, these properties modulate the nanoceria’s behavior in different environments and thus its distribution, elimination and toxicity, at both the cellular level and in the body.

In this study, we have investigated for the first time the consequences of the exposure to CeO_2_ NPs of primary trophoblasts from human placenta at term. In order to compare our results with those of other publications, we used here a wide range of concentrations, from the lowest concentrations (0.6 µg/mL, corresponding to 100 ng/cm^2^) that still allow their detection by classical methods, to a very high concentration (640 µg/mL or around 100 µg/cm^2^) corresponding to high levels of atmospheric pollution (to compare results with those from pulmonary exposures) (Table 1). We focused our study on the uptake of CeO_2_ NPs by human primary trophoblast cells, their potential cytotoxicity and mechanisms of action as regard to the induction of oxidative stress, anti-oxidant enzymes and caspase activation. We explored the impact of CeO_2_ NPs on the ability of trophoblasts to differentiate and form the syncytiotrophoblast in vitro, which is in direct contact with maternal blood and represents the endocrine tissue, and evaluated the main genes that regulate this process. Last, we determined the endocrine function of placenta by measuring hormone secretion and the levels of the main genes and enzymes involved in hormone production and steroidogenesis. Altogether, our data shed new light on the hazards of ceria NPs in human placental trophoblast cells.

## 2. Materials and Methods

### 2.1. Placenta Collection

Placentas were collected from non-smoking, healthy women with pregnancies delivered by Caesarean sections between 37 and 39 weeks of amenorrhea (hereafter called ”term”). Placentas were obtained from the Port-Royal Maternity, the Mutualist Institute Montsouris, the Private Hospital of Antony, the Béclère Hospital, and the Beaujon Hospital after obtaining written consent from informed patients and approval from our local ethics committee (CPP: 2015-May-13909). The study was performed according to the principles of the Declaration of Helsinki.

### 2.2. Cytotrophoblast Purification, Culture and Treatment

After collection, placental tissues were washed in Ca^2+^- and Mg^2+^-free Hank’s Balanced Salt solution (HBSS, Gibco #14175, Thermo Fisher Scientific, Illkirch, France). Then, chorionic villi were gently scraped free from vessels and connective tissue, and dissected into about 25 mg fragments. The time for placental dissection was kept under 30 min to prevent tissue degradation. The mononucleated villous cytotrophoblasts (VCT) were isolated, based on the methods of Kliman et al. [31]. After dissection, the chorionic villi were washed in Ca^2+^- and Mg^2+^-free HBSS, and then digested in trypsin digestion medium containing HBSS 5 mL/g, 0.1% trypsin (Sigma-Aldrich #27250-018, Saint Quentin Fallavier, France), 0.1 M MgSO_4_ (Merck #5886-0500, Fontenay sous Bois, France), 0.1 M CaCl_2_ (Merck #1-02820-1000), 4% milk and 50 Kunitz/mL DNAse type IV (Sigma-Aldrich #D5025, Saint Quentin Fallavier, France), for 30 min at 37 °C without agitation. This was repeated eight times with 10 min incubation with the same trypsin solution. The first three trypsin digestions, containing a mixture of extravillous cytotrophoblasts (EVCT) and VCT, were discarded after light microscopic analysis, and the last three ones, containing a majority of VCT, were kept and pooled. The chorionic villi were finally washed with warm HBSS (37 °C). Each time, the supernatant-containing VCT was collected after tissue sedimentation, filtered (on 40-µm pores filters), and incubated with 10% fetal calf serum (FCS, vol/vol) to stop trypsin activity. After purification by Percoll gradient, VCT were resuspended and cultured in DMEM (containing 1 g/L glucose, pyruvate, Phenol red-free, Thermo Fisher Scientific, #11880, Illkirch, France) supplemented with 10% FCS (Eurobio #CVFSVF00-01, Les Ulis, France), 2 mM glutamine (Sigma-Aldrich #G7513), 100 IU/mL penicillin and 100 µg/mL streptomycin (Gibco #15140-122, Thermo Fisher Scientific) at 125,000 cells/cm^2^ on 35-mm or 60-mm diameter culture dishes. Trophoblast cells were seeded at a density of 125,000 cells/cm^2^ in 24-, 48-, 96-wells or in 35-mm or 60 mm plates (Techno Plastic Products (TPP), Trasadingen, Switzerland). After around 16 h of culture (overnight), VCT were carefully washed to eliminate non-adherent cells. Purified VCT were characterized for each culture to ensure the homogeneity between purifications by microscopic visualization, the ability to aggregate at 48 h of culture and to form ST at 72 h, and by monitoring the production of hCG secreted into the supernatant. Cells were collected at 24 h, 48 h and 72 h, kept at −80 °C for total RNA or protein extraction, fixed in cold methanol for further immunolabeling, or fixed in 2% paraformaldehyde/2.5% glutaraldehyde for TEM analysis. Staurosporine (Stp, Sigma-Aldrich #S4400) was dissolved in DMSO at 2 mM and stored at −20 °C. Trophoblasts were incubated for 4 hours with Stp at a final concentration of 0.5 μM. Tert-butyl hydroperoxide stock solution (TBHP, Luperox TBH70X solution, Sigma-Aldrich #458139) was stored at 4 °C. Trophoblasts were incubated from 1 to 3 h with TBHP at final concentrations between 100 and 250 μM. The NP concentrations used in the study are expressed in μg/mL (aiming to allow comparison to most of the NP publications on placenta models). Taking into account the volume of culture medium according to the different areas of the culture dishes, the corresponding concentrations in μg/cm^2^ are presented in Table 1.

### 2.3. Cerium Dioxide Nanoparticle Preparation and Characterization

CeO_2_ NPs were obtained from the Joint Research Center of the European Union (NM-212, IHCP, Ispra, Italy). NPs were dispersed in DMEM cell culture media without FCS at a concentration of 3 mg/mL through sonication, with a sonifier equipped with a cup horn (450 W and 50/60 Hz, Branson, Danbury, USA) at 70% amplitude, on ice, for 2 min. NPs were then sequentially diluted in cell culture medium with FCS immediately before use, to give a final concentration range of 0.6–640 µg/mL. Dynamic light scattering and zeta-potential determinations were performed in water, in cell culture media without FCS and in cell culture media with FCS, with the Zetasizer Nano ZS (Malvern Instruments, Worcestershire, UK). DLS measurements were performed in triplicate with a particle concentration of 0.05 mg/mL at 37 °C, at a scattering angle of 173°. Zeta-potential measurements were performed in triplicate at 37 °C. The zeta-potential was calculated by fitting the electrophoretic mobility with the Smoluchowski model. For these Zeta-potential measurements, culture media with or without FCS were diluted in water (1:10).

### 2.4. Cell Viability by WST1

Cell viability was assessed on the basis of mitochondrial metabolic activity of cells using the WST1 assay (Sigma-Aldrich #11644807001)—in which mitochondria dehydrogenase cleaves the tetrazolium salt 2-(4-idophenyl)-3-(4-nitrophenyl)5-(2,4-disulphophenyl)-2H-tetrazolium (WST1) into formazan—and according to the manufacturer’s instructions. Trophoblast cells were cultured overnight in 48-well plates, washed with fresh medium and treated accordingly for 24 to 72 h. At the end of the exposure time, cells were rinsed with culture medium and then WST1 reagent was added (1:100) to each well and incubated for 3 h at 37 °C. Spectrometric absorbance was measured using a microplate reader (EnSpire 2300 Multilabel reader, PerkinElmer, Villebon-sur-Yvette, France) at 440 nm, and using 600 nm as the correction wavelength (to remove the non-specific emission). To measure background noise from NPs, we performed the same experiment in parallel, followed by the addition of Triton to each well 15 min before adding WST1 reagent, and the corresponding absorbances were subtracted from the results.

### 2.5. Cytotoxicity by LDH Release

Cytoplasmic membrane permeability was evaluated by assaying lactate dehydrogenase (LDH) in the culture supernatants. After treatment, the culture supernatants were kept and centrifuged for 3 min at 350× *g* to eliminate cells, debris and NPs. LDH concentrations were determined using a Roche/Hitachi cobas c 701/702 analyzer with a detection range of 10–1,000 IU/L. Within-run and between-run imprecision (CV%) values were <2.7%.

### 2.6. Caspase Activation Assay with ApoONE^®^

The induction of apoptosis was assessed by the estimation of the activity of caspases 3 and 7 using the Apo-ONE Homogeneous Caspase-3/7 assay (Promega #G7791, Charbonnières-les-Bains, France) according to the manufacturer’s instructions. Trophoblast cells were cultured in a 96-well plate and treated accordingly. At the end of treatments, the mixture of reagents was added to each well directly in the cell media and incubated for 1 h at room temperature. Fluorescence intensity was measured at excitation/emission wavelengths of 499 nm/521 nm using a microplate reader (EnSpire 2300 Multilabel reader, PerkinElmer, Villebon-sur-Yvette, France). Interference of NPs with the fluorescence intensity was determined by measurements of plates before and immediately after adding NPs to the wells of staurosporine-treated cells. Thus, we determined that the particles themselves did not interfere with the florescence intensity when used at concentrations of 2.5, 10 and 40 μg/mL, unlike the concentration of 320 μg/mL, which reduced fluorescence intensity by 30%.

### 2.7. Reactive Oxygen Species (ROS) Detection by CM-H_2_DCFDA

Reactive oxygen species (ROS) levels were determined by measuring the oxidation of the fluorescent probe 5′,6′-chloromethyl-2′,7′-dichlorodihydrofluorescein diacetate (CM-H_2_DCFDA, Thermo Fisher Scientific #C6827). Trophoblast cells seeded in a 96-well plate were washed with PBS and incubated with 10 μM CM-H_2_DCFDA for 45 min at 37 °C, washed again with PBS and then incubated with or without NPs or TBHP for 4 to 72h. Fluorescence intensity was measured at excitation/emission wavelengths of 485/520 nm using a microplate reader (EnSpire 2300 Multilabel reader, PerkinElmer).

### 2.8. Assays of Hormone Secretions

hCG, progesterone (P4), estradiol (E2), hPL and PlGF were assayed in the culture supernatants of ST at 72 h of culture. At the end of treatments, supernatants were centrifuged for 3 min at 350× *g* to eliminate cell debris. Total hCG and P4 concentrations were determined using ECLIA immunoassay (Liaison^®^, DiaSorin[TS1]) with a detection range of 1.5–10,000 IU/L and 0.7–190 nmol/L, respectively. Within-run and between-run imprecision (CV%) values were below 5% and 11%, respectively; Detection limits were 0.3 IU/L and 0.4 nmol/L, respectively. E2 concentration was determined using ECLIA immunoassay (Liaison^®^, Estradiol II Gen) with a detection range of 10–1000 pg/mL. Within-run and between-run imprecision (CV%) values were below 3.9% and to 10.7%, respectively. The detection limit was 16.2 pg/mL. hPL concentrations were determined using an ELISA kit (DiaSource #KPAD1283, Ottignies-Louvain-la-Neuve, Belgium) with a detection limit of 0.04 mg/L. Supernatants of primary culture were diluted with the diluent provided (1:10) and assayed in duplicate. Optical densities were read at 450 nm on a microplate reader (EnSpire 2300 Multilabel reader, PerkinElmer). PlGF concentrations were determined using an ELISA kit (R&D Systems, #DPG00) with a detection limit of 7.0 pg/mL. Supernatants of primary culture were diluted with the RD5K diluent (1:10) provided and assayed in duplicate. Optical densities were read at 450 nm and 540 nm for the corrected wavelength on a microplate reader (EnSpire 2300 Multilabel reader, PerkinElmer).

Interferences of NPs with hormones (hormones adsorbed on NPs that are potentially discarded from samples by centrifugation) was determined by acellular incubation of supernatants after 72 h of culture of control cells, with three concentrations of NPs used in this study (2.5, 40 and 320 μg/mL) for 4 h at 37 °C, followed by a centrifugation step for 3 min at 350× *g*. Then, all assays were performed as previously and the results were compared to those of the same supernatant of control cells in the absence of acellular incubation with NPs. We concluded that NP reduced the results: by 7% for hCG whatever the concentration, by 20% for LDH only at 320 μg/mL (50.4 μg/cm^2^), by 15–20% for P4 whatever the concentration, and by 10% for E2 & PlGF only at 320 μg/mL (50.4 μg/cm^2^).

### 2.9. RNA Extraction

Trophoblast cells cultured in 35-mm culture dishes were lysed in TRIzol^®^ (Tri Reagent^®^, Sigma-Aldrich #T9424). Total RNAs were then extracted in accordance with the manufacturer’s instructions using the Direct-zol RNA Miniprep kit (Zymo research, #R2052, Saint-Quentin-en-Yvelines, France), with DNase treatment as recommended. RNA concentrations were determined with a NanoDrop (NanoDrop ONE, Thermo Fisher Scientific) spectrophotometer by measuring optical density at 260 nm. The quality of RNAs was considered suitable for use when both A260/A280 and A260/A230 ratios were between 1.8 and 2.

### 2.10. Reverse Transcription and Quantitative PCR (RT-qPCR)

Reverse transcription was performed using the high-capacity cDNA reverse transcription kit (Applied Biosystems #4368814, Villebon sur Yvette, France) with 500 ng of RNA. RNAs were first linearized at 65 °C for 5 min and in the presence of 0.1 μg random primers (Invitrogen, #48190011) and 10 mM dNTP (Invitrogen, #10297018) in a 13 μL reaction volume. Reverse transcriptions were performed in a 20 μL final reaction volume at 50 °C for 50 min and at 70 °C for 15 min. Gene-specific primers used for real-time PCR were designed using the OLIGO Explorer software (Molecular Biology Insights, Hawthorne, NY, USA) (Table 2). Quantitative real-time PCR was carried out in a 10 μL reaction volume containing 500 ng of cDNA, 45 nM of each primer, and Takyon™ MasterMix (Takyon™ Rox SYBR^®^ MasterMix dTTP Blue, Eurogentec, Angers, France) using a CFX384 Real-Time System for the detector coupled with a C1000 Touch Thermal Cycler (Bio-Rad). PCR cycles consisted of the following steps: Takyon activation (3 min, 95 °C), denaturation (10 s, 95 °C), and annealing and extension (1 min, 60 °C). The threshold cycle (Cq) was measured as the number of cycles at which the reporter fluorescent emission first exceeds the background. Data were analyzed with the CFX Maestro software (Bio-Rad). The relative amounts of mRNA were estimated using the regression method and then expressed as fold change. Primers for *HPRT*, *SDHA RPL13* and *RPLO* were used as reference genes for the normalization of the results. The results are given as fold change and each gene was normalized to the geometric mean of reference genes that were found to be unchanged among the different conditions used.

### 2.11. Western Blot

Total protein extracts from trophoblast cells from 60-mm dish cultures were obtained by harvesting cells in Laemmli buffer (0.06 M Tris-HCl, pH 6.8, 10% glycerol, 2% SDS, protease inhibitors (Protease Inhibitor Cocktail Set I, Calbiochem Merck #539131), phosphatase inhibitors (Phosphatase Inhibitor Cocktail Set V, Merck #524629)). Protein extracts were centrifuged for 10 min at 14,000× *g* to eliminate NPs and cell debris, heated for 10 min at 70 °C, and then sonicated. Protein concentrations were determined using the Pierce™ BCA Protein Assay Kit (Thermo Fisher Scientific #23235). Equal amounts of proteins (30 µg) were separated on 4–15% or 8–16% SDS-PAGE mini-PROTEAN^®^ TGX™ precast protein gel (Bio-Rad #4561084 #4561103, Marnes-la-Coquette, France) under reducing conditions (DTT 10%, Sample Reducing Agent 10X, Invitrogen #NP0009) and transferred onto a nitrocellulose membrane (Trans-blot Turbo Transfer pack, 0.2 µm nitrocellulose, Bio-Rad #1704159). Blots were incubated overnight with the primary antibody at 4 °C, and then for 1 h with the appropriate DyLight 680 or 800 Fluor-conjugated secondary antibody (Thermo Fisher Scientific #35568 and #35521). The primary antibodies used were: mouse monoclonal anti-β-actin (A5441, Sigma-Aldrich, 2.6 μg/mL), rabbit polyclonal anti-aromatase (CYP19A1, 0.3 μg/mL, Abcam #ab66981), rabbit monoclonal anti-catalase (Cell Signaling Technology #12980, Leiden, The Netherlands), rabbit polyclonal anti-heme oxygenase 1 (HO1, 1 μg/mL, Enzo Life Sciences #ADI-SPA-895, Villeurbanne, France), rabbit monoclonal anti-HSD3β1 (0.18 μg/mL, Abcam #ab167417, Cambridge, UK), rabbit monoclonal anti-HSD17β1 (0.0152 μg/mL, Abcam #ab51045), rabbit polyclonal anti-MLN64 (1 μg/mL, Abcam #ab3478), mouse monoclonal anti-p53 (1:1000, Clone DO-7, DakoCytomation #M7001), rabbit polyclonal anti-P450scc (1 μg/mL, Abcam #ab75497), rabbit polyclonal anti-PARP1 (1:1000, Cell Signaling Technology #9542), rabbit polyclonal anti-Prx-SO_3_ (1:1000, Abcam #ab16830), mouse monoclonal anti-SOD1 (1:1000, Cell Signaling Technology #4266), rabbit monoclonal anti-SOD2 (1:1000, Cell Signaling Technology #13141), rabbit polyclonal anti-STS (1:1000, Abcam #ab62219), mouse monoclonal anti-Vinculin (1 μg/mL, V9131, Sigma-Aldrich). Secondary antibodies were DyLight 800-labeled anti-mouse (Thermo Fisher Scientific #35521) or DyLight 680-labeled anti-rabbit (Thermo Fisher Scientific #35568), and blots were scanned with an Odyssey^®^ Imaging System (Li-COR, Bad Homburg, Germany). Quantification was performed using Li-COR Odyssey software.

### 2.12. Cell Immunolabeling and Fusion Index

After trophoblast culture in 35-mm culture dishes, cells were washed with PBS, fixed and permeabilized in cold methanol for 8 min, and saturated with a solution of 1% BSA and 0.1% Tween in PBS for 2 h at room temperature. The primary antibody, anti-desmoplakin (5 μg/mL, Abcam #ab71690), was prepared in PBS containing 1% BSA and 0.1% Tween and incubated overnight at 4 °C. After washing, the cells were incubated with Alexa Fluor 546 donkey anti-rabbit antibody (4 μg/mL, Invitrogen #A11010, Illkirch, France) diluted in PBS containing 1% BSA and 0.1% Tween for 2 h at room temperature and protected from light. Nuclei were stained with 4′,6-diamidino-2-phenylindole (DAPI) for 5 min at room temperature and protected from light, and then mounted on glass slides with Dako fluorescence mounting medium (Dako #S3023, Les Ulis, France) for microscopy. Images were taken by epifluorescence microscope (BX60, Olympus, Rungis, France), equipped with a 40x oil objective (Olympus 1.00), an ultrahigh-vacuum mercury lamp and a Hamamatsu camera. Pictures were taken with VisionStage Orca Software (v1.6) associated with the Hamamatsu camera. Syncytium formation was followed by monitoring the cellular distribution of desmoplakin and nuclei, termed the fusion index. The fusion index was calculated using the previously published method [32,33], from nine non-overlapping images per replicate and determined as [(N−S)/T], where N equals the number of nuclei in syncytia, S equals the number of syncytia, and T equals the total number of nuclei counted (VCT plus ST).

### 2.13. Transmission Electronic Microscopy (TEM)

Cells were seeded in 60-mm culture dishes and treated adequately. Cells were collected by trypsinization and centrifuged for 5 min at 1400× *g*. The cells were resuspended and washed in PB buffer (0.05 M PIPES [piperazine-N,N′-bis(2-ethanesulfonic acid)], 5 mM CaCl_2_, pH 7.3) for 10 min, then centrifuged for 5 min at 1400× *g* and fixed in PB containing 2.5% glutaraldehyde and 2% paraformaldehyde for 45 min at room temperature. After 5 min of centrifugation at 1400× *g*, cells were washed twice for 10 min in PB. The samples were post-fixed for 45 min in PB containing 1% osmium tetroxide at 4 °C and then in 1% aqueous uranyl acetate solution for 2 h at room temperature. The samples were then dehydrated in serial graded ethanol solutions (30%, 50%, 70%, 95% and 100%, 3 times for 10 min each) followed by ethanol/propylene oxide (1/1 [vol/vol]) and propylene oxide (10 min) and finally in pure propylene oxide (3 times for 10 min). Each sample was finally embedded in Epon epoxy resin. Ultrathin sections (of 80 nm thickness for TEM and of 200 nm for EDX) were cut with a Leica ultracut S microtome fitted with a diamond knife (Diatome histoknife Jumbo or Diatome ultrathin), stained with 2% uranyl acetate and lead citrate and transferred onto copper grids. The microscope used for the study was a Jeol electron microscope (JEM-100S, Croisy sur Seine, France) at 80 kV with an Orius SC200 digital camera (Gatan-Roper Scientific, Evry, France). Acquisitions were performed with Gatan software. From the micrographs, the sizes of intracellular NP aggregates were measured with ImageJ Software using the Analyse Particles module on 50 morphologically preserved cells containing NPs, chosen randomly in each condition. The percentage of cells containing NPs was determined by observation of 200 morphologically preserved cells. Cells displaying morphological signs of cellular death or whose nuclei were not clearly perceptible were systematically excluded.

### 2.14. Scanning TEM and EDX Analysis

For energy dispersive X-ray (EDX) and scanning transmission electronic microscopy (STEM)-EDX analysis, cell sections (200 nm) were cut with an ultramicrotome EM UC6 (Leica Microsystems, Nanterre, France) and, respectively, collected on glass slides or formvar carbon-coated copper grids (Agar Scientific Ltd., Stansted, Essex, UK). EDX studies were performed with a JEOL (JEM-1400) transmission electron microscope operating at 120 kV. The microscope was equipped with a SAMx EDS SD-Detector in order to obtain information on the chemical composition of the cell sections. Spectral data were acquired with IDEFix software for 60 s at 10,000 magnification, and the sections were tilted between 10° toward the detector. Cerium signal was detected at 4.84 keV (Ce Lα1). Dark-field STEM images were collected with the annular dark field detector with 8 cm camera length. EDX chemical maps in STEM mode were recorded for 30 min with Maxview software.

### 2.15. Statistical Analysis

All experiments were reproduced with at least three independent placentas and the results of quantitative analysis are presented as means +/− SEM. Differences between groups were evaluated with the paired Student’s t-test using GraphPad Prism 6.0 software. The level of significance was fixed at a *p*-value < 0.05 (*), < 0.01 (**), < 0.001 (***) or < 0.0001 (****).

## 3. Results

### 3.1. NP Characterization

Before any cell culture experimentation, the behavior of CeO_2_ NPs was characterized in different media. Zeta potential and hydrodynamic diameter of the suspended particles in water and in diluted DMEM cell culture media with (w/) or without (w/o) fetal calf serum (FCS) are summarized in Table 3, which also summarizes physicochemical data from the provider on primary particle size (SEM), BET (Brunauer, Emmett and Teller method) surface area and crystalline structure of the CeO_2_ NPs used. The state of agglomeration of NPs, directly or 1h after sonication, was determined by dynamic light scattering (DLS) and the size distribution is shown in Figure 1. The DLS results show that NPs were monodispersed and the distribution curves show variable degrees of agglomeration of CeO_2_ NPs in all the media tested, with larger agglomerates in DMEM with FCS. The average hydrodynamic diameters of CeO_2_ NPs in water, in DMEM without FCS and in DMEM with FCS were 378 ± 65 nm, 441 ± 57 nm and 503 ± 55 nm, respectively, directly after sonication (Figure 1A and Table 3). We performed the same measurements 1 h after sonication and the hydrodynamic diameters were roughly the same with respective values of 441 ± 43 nm, 411 ± 51 nm and 443 ± 80 nm in the same solutions (Figure 1B and Table 3). Thus, the average hydrodynamic sizes were much larger than their primary size determined by the manufacturer (28.4 ± 10.4 nm) (Table 3), suggesting that CeO_2_ NPs might easily form agglomerates and/or aggregates in suspension. This aggregation increased after 24 h but remained stable for longer incubation periods and was greater in culture media compared to suspensions in water (high NP aggregation was observed by DLS in DMEM at 24 h, 48 h and 72 h). The zeta potential analysis showed that CeO_2_ NPs have a sharp negative surface charge when suspended in water with a value around −32.7 ± 0.9 mV.

Due to the high content of salts, the CeO_2_ charge was measured in diluted DMEM medium (1 DMEM: 10 H_2_O [*v/v*]). In both DMEM physiological solutions without or with FCS, the surface charges were still negative but with lower values (−21.7 ± 2.8 and −23.4 ± 1.0 mV, respectively) (Table 3).

### 3.2. Internalization of Cerium Dioxide Nanoparticles in Human Trophoblasts

To determine whether human trophoblasts adsorb NPs at their cell surface or can even internalize CeO_2_ NPs, we performed transmission electron microscopy (TEM) (Figure 2). To this end, human VCT were purified from placenta at term of pregnancy (37–39 weeks of amenorrhea (WA)), were left to plate overnight and were either treated as VCT, at 24 h of culture (Figure 2A), or left in culture for 3 days to spontaneously differentiate into ST before incubation with CeO_2_ NPs (Figure 2B). For the concentrations of NPs, we tested a range from low to relatively high levels (0.6, 2.5, 10 and 40 μg/mL corresponding to 0.1, 0.4, 1.6 and 6.3 μg/cm^2^, respectively (Table 1)) for 24 h of incubation. Figure 2A shows that NPs are internalized by VCT regardless of the concentrations tested and that intracellular NPs are assembled in the cytosol, mostly in a perinuclear zone, where they form agglomerates/aggregates. Even after deep exploration of the images from TEM micrographs, we did not detect any NPs in the cellular organelles (e.g., the nucleus, mitochondria, the Golgi apparatus or endoplasmic reticulum). However, NPs could often be visualized in vesicle-like structures within the cytosol (Figure 2A, higher magnification). After exposure to CeO_2_ NPs at these concentrations, the majority of VCT cells were morphologically preserved, with an intact cytoplasmic membrane and nucleus, as visualized by toluidine staining (Appendix A). We then estimated the proportion of cells that internalized NPs as a function of the concentration used by randomly counting 200 cells for each condition (Figure 2C). As expected, the internalization of NPs is concentration-dependent and ranged from approximately 3% of the cells containing NPs at the concentration of 0.6 μg/mL (0.1 μg/cm^2^) to more than half (58%) of the cells containing NPs at the concentration of 40 μg/mL (6.3 μg/cm^2^). The sizes of the intracellular agglomerates of NPs were measured in 50 cells for each condition and the size distribution is shown in Figure 2D (upper bar graph). For each concentration, we found that the NPs occur mostly as individual NPs (around 40% between 140 and 1000 nm^2^, corresponding to a size of 10–30 nm), even though they appear grouped in the cytosol, and that the number of agglomerates decreases with increasing size (6 to 9 % above 10^5^ nm^2^, corresponding to a size above 316 nm). Next, we did the same experiments on terminal differentiated trophoblasts, the ST cells, which are physiologically more relevant as they are the tissue in direct contact with the maternal blood, and thus with pollutants. The ST can be easily identified as their nuclei appear lighter under TEM and present chromatin condensation spots (because of the chromatin remodeling of these differentiated cells), and harbor several clumped nuclei (Appendix A and Figure 2B). NPs were internalized in ST in the same way as in VCT (Figure 2B). ST uptake of NPs was also concentration-dependent and ranged from 6% of the cells containing NPs at a concentration of 0.6 μg/mL (0.1 μg/cm^2^) to more than half (56%) of the cells containing NPs at a concentration of 40 μg/mL (6.3 μg/cm^2^) (Figure 2C). To compare the capacity of ST versus VCT to accumulate NPs, we evaluated the size of NPs agglomerates in ST (Figure 2D, lower bar graph). The size distribution was nearly the same as in VCT, with a higher number of individual nanoparticles (around 35% between 140 and 1000 nm^2^).

Next, to confirm that the agglomerates that we visualized in trophoblasts under TEM are of CeO_2_ NPs, we performed energy dispersive X-ray (EDX) analysis by TEM of cells incubated with 40 μg/mL (6.3 μg/cm^2^) of CeO_2_ NPs and compared them with untreated cells (Figure 3). The EDX analysis determines the chemical composition of the cell sections. In the micrographs, we found the energy peak (4.839 keV) corresponding to cerium (Ce) in cells incubated with NPs (Figure 3A, right graph and the corresponding TEM image) and their absence from control cells (Figure 3A, left graph and the corresponding TEM image). We next established the chemical mapping of control cells and of NP-treated cell sections by scanning transmission electron microscopy (STEM) dark field and EDX analysis. The Dark field STEM images show the Z-contrast according to the scattering of electrons, and EDX gives the cartography, pixel by pixel, of the chemical element of interest present in the sample. In Figure 3B, we detected Ce in co-localization with the NP cluster visible in the STEM and dark field images (Figure 3B, right images). Although some dispersed background pixels of Ce were detected, there was no agglomerate of Ce visualized in the control cells (Figure 3B, left images). Taken together, these results demonstrate that human primary trophoblasts, both VCT and fully differentiated ST, are able to internalize CeO_2_ NPs in a concentration-dependent manner.

### 3.3. Impact of Cerium Dioxide Nanoparticles on Human Primary Trophoblast Cell Viability

We next investigated the effects of NPs on cellular viability (Figure 4). To determine whether NPs have cytotoxic effects on trophoblasts, VCT were exposed to a range of concentrations of CeO_2_ NPs from 2.5 to 640 μg/mL (corresponding to 0.4 to 101 μg/cm^2^ (Table 1)) for 24 h to 72 h. WST1 colorimetric cell viability assays were performed after each incubation period (Figure 4A). High concentrations of CeO_2_ NPs (above 80 μg/mL (12.6 μg/cm^2^)) decreased trophoblast cell viability after 24 h-exposure, and from a much lower concentration (5 µg/mL (0.8 μg/cm^2^)) at a longer exposure time (48 h incubation). However, WST1 is a metabolic test that takes into account changes in mitochondrial activities by pollutants. Thus, in parallel, release of LDH, an intracellular glycolytic enzyme, was quantified in supernatants of trophoblasts exposed to CeO_2_ (Figure 4B). The amount of LDH release in cell media reflects cell membrane integrity loss and thus is a sensitive marker for cellular toxicity. A significant increase of LDH release was observed only at extremely high concentrations of CeO_2_ NPs (320 μg/mL (50.4 μg/cm^2^)) for all incubation times.

Moreover, we evaluated the activation of caspases 3 and 7, two apoptosis effectors, using a caspase activity test after a 24 h-exposure to four concentrations of CeO_2_ NPs (2.5, 10, 40 and 320 µg/mL, corresponding to 0.4, 1.6, 6.3 and 50.4, respectively) and compared to incubation with staurosporine (Stp), a classic apoptosis inducer (Figure 4C). A very low but significant activation of these caspases was induced by the concentrations higher than 10 µg/mL (1.6 μg/cm^2^) of NP (1.25 ± 0.08, 1.55 ± 0.19 and 2.21 ± 0.24 fold increase for 10, 40 and 320 µg/mL, respectively, versus 7.13 ± 0.94 for Stp) suggesting that some of the cells may enter the apoptotic process. The results were underestimated for the concentration of 320 µg/mL (50.4 μg/cm^2^) of NPs, because the test of NP interference for each concentration used showed only a decrease (30%) of the fluorescence at 320 µg/mL, but this was not the case for the lower concentrations tested. We also evaluated the level of p53 protein and of PARP-1 by Western blot (Figure 4D). The apoptosis inducer p53 is stabilized at the protein level in response to DNA damage or oxidative stress and its activation leads to either cell cycle arrest or commitment to apoptosis. PARP-1, a nuclear poly (ADP-ribose) polymerase involved in DNA repair, is a caspase-3 direct target whose cleavage (cleaved PARP-1) serves as a marker of cells undergoing apoptosis. Quantifications of Western blots do not show major variations in p53 level even at the highest concentrations of CeO_2_ used, although a very low but significant stabilization of p53 was obtained with the lowest concentration of 2.5 µg/mL (0.4 μg/cm^2^) of CeO_2_ NPs (Figure 4D, left quantification graph). PARP-1 cleavage was only detected with high concentrations of NP (320 µg/mL (50.4 μg/cm^2^)), with a significant increase of almost the same extent as in Stp-positive controls (Figure 4D, right quantification graph). Altogether, our results indicate that CeO_2_ NPs are toxic for human primary trophoblasts at extremely high concentrations and that this involves caspase activation in the absence of p53 stabilization, although much lower concentrations can affect the metabolic activity of the trophoblasts after longer times of exposure.

### 3.4. Impact of Cerium Dioxide Nanoparticles on the Oxidative State of Human Trophoblasts

We investigated the effects of CeO_2_ NPs on the cellular oxidative stress of human trophoblasts (Figure 5) by measuring the production of ROS using the CM-H_2_DCFDA probe after exposure of primary trophoblasts to increased concentrations of CeO_2_ NPs (from 2.5 to 320 µg/mL, corresponding to 0.4–50.4 μg/cm^2^) for 4 h (Figure 5A), 24 h, 48 h or 72 h (Appendix A). The stabilized H_2_O_2_ tert-butyl hydroperoxide (TBHP), described as a strong oxidative stress inducer, was used here as a positive control for each time point, and showed that the trophoblasts are sensitive to H_2_O_2_, as reported before [34]. As shown in Figure 2A, there was a slight increase in ROS production when VCT were incubated with CeO_2_ NPs at 5 to 20 µg/mL (0.8 to 3.2 μg/cm^2^) for 4 h, but this was no longer the case after 24, 48 or 72 h of incubation (Appendix A). On the contrary, there was a decrease in ROS production at concentrations of CeO_2_ NPs higher than 160 μg/mL (50.4 μg/cm^2^) at 4 h of incubation. This profile was similar at a longer time of exposure (24 h, 48 h or 72 h) with a marked decrease in ROS production at concentrations of CeO_2_ NPs higher than 80 μg/mL (12.6 μg/cm^2^). To determine if the exposure to CeO_2_ NPs triggered an adaptive antioxidant response, protein levels of the main antioxidant enzymes were evaluated by Western blot, such as: SOD1, SOD2, catalase, HO-1, and sulfinic 2-Cys peroxiredoxines (Prx-SO_3_), a sensor of intracellular H_2_O_2_ (Figure 5B and Appendix A). As shown with the positive control TBHP, trophoblasts were able to develop an antioxidant response (at 4 h or 24 h), confirmed by the increase of SOD1, SOD2 and Prx-SO_3_ expressions. After cytotrophoblast exposure to NPs, there were no significant modifications of these antioxidant enzymes and sensors with any of the concentrations of NPs tested after 4 h or 24 h of incubation, except for SOD1 with a slight decrease at 10 and 40 μg/mL (1.6 and 6.3 μg/cm^2^, respectively) (Figure 5B and Appendix A). Together with the lack of stabilization of the p53 protein shown in Figure 4, these results show that CeO_2_ NPs did not activate antioxidant response pathways at the highest concentrations used.

### 3.5. Effects of Cerium Dioxide Nanoparticles on Human Primary Trophoblast Differentiation

The differentiation of human primary VCT is characterized by syncytialization, their capacity in vitro to spontaneously aggregate and fuse to form the syncytiotrophoblast, concomitant with an increase in hormonal secretion by the syncytiotrophoblast. We have investigated the effect of two concentrations of CeO_2_ NPs, a low concentration (2.5 µg/mL (0.4 μg/cm^2^)) and a relatively high concentration (40 µg/mL (6.3 μg/cm^2^)), which affect metabolic activity without LDH release, on the differentiation of human VCT into ST in vitro. The fusion index was calculated using a published method [32,33] based on the immunostaining of desmoplakin, a marker of the cellular boundaries (component of desmosome), and of nuclei staining with DAPI (Figure 6). Syncytialization leads to the formation of multinucleated cells, distinguished by several nuclei surrounded by a single plasma membrane. In comparison to untreated VCT, the graph shows a 20% decrease in syncytium formation for the two concentrations of NP tested, with a fusion index of 0.83 ± 0.02 for NPs at 2.5 µg/mL (0.4 μg/cm^2^) and 0.81 ± 0.04 for NPs at 40 µg/mL (6.3 μg/cm^2^) (Figure 6, right graph). When we analyzed the expression of the main actors of VCT differentiation (syncytin 1, syncytin 2, GCM1 and CDH1) and of syncytin-specific receptors (ASCT2 and MFSD2) by qPCR using mRNA extracted from untreated VCT and VCT exposed to CeO_2_ for 24 h, no major changes were found whatever the concentration of NPs used (2.5, 10 and 40 µg/mL, corresponding to 0.4, 1.6 and 6.3 μg/cm^2^, respectively) (Appendix A). However, because of the lack of reliable antibodies, we could not draw conclusions regarding their protein levels. Fusion index results indicate that CeO_2_ NPs markedly affect the capacity of trophoblasts to form a ST, with no deregulation in expression of the master participants involved in the formation of syncytiotrophoblasts.

### 3.6. Effects of Cerium Dioxide Nanoparticles on the Endocrine Function of the Syncytiotrophoblasts

We next investigated the impact of NPs on hormonal function of the ST. Hormones secreted by the ST include polypeptide hormones like human chorionic gonadotropin (hCG) and placental lactogen (hPL), and steroid hormones such as progesterone (P4) and estrogen (E2). The trophoblast also produces placenta growth factor (PlGF), a ligand of the VEGF receptor 1 (VEGF-R1), which promotes angiogenesis, and sFlt-1, the soluble form of VEGF-R1 with antiangiogenic properties. The fine balance of pro- and anti-angiogenic factors is often deregulated in several pregnancy disorders such as preeclampsia [35,36]. Levels of peptide hormones (hCG and hPL), PlGF angiogenic factor and steroid hormones, like progesterone (P4) and estradiol (E2), which are released in the culture media by trophoblasts after 72 h of culture, were analyzed from the same samples depicted in Figure 4A,B (Figure 7A). Interferences of NPs with hormones were determined as described in Methods. In Figure 7, we show that at 40 µg/mL (6.3 μg/cm^2^) (a concentration of NP that showed decreased metabolic activity of culture without LDH release at 72h), the CeO_2_ NPs modify the secretion of hCG and hPL, although considerable inter-individual placenta variations are observed at lower concentrations, but without statistical significance for the pool of n = 9 placentas. In the case of PlGF secretion, CeO_2_ NPs induced no effect at concentrations from 2.5 to 40 μg/mL (0.4 to 6.3 μg/cm^2^). A marked decrease of hCG, hPL and PlGF secretions was observed with the highly toxic concentration of 320 μg/mL (50.4 μg/cm^2^), and this follows the course of cell metabolic activity (blue bars showing average values of WST1 assay), thus this decrease is due to the mortality of the trophoblasts. When we evaluated gene expression of hCGβ, hPL, PlGF and sFlt1 by qPCR after 24 h of exposure to NPs (Appendix A), we observed no significant variation at the concentrations tested (2.5, 10 and 40 μg/mL, corresponding to 0.4, 1.6 and 6.3 μg/cm^2^, respectively). Among the steroids, the secretion of P4 was maintained at the level of untreated trophoblasts despite the drastic decrease in cell viability (38 ± 7% living cells) with the highly cytotoxic concentrations of CeO_2_ NPs (320 µg/mL (50.4 μg/cm^2^)). Moreover, lower concentrations of NPs tended to perturb P4 secretion, with a higher production of P4 relative to the metabolically active cells and in a concentration-dependent manner (with a ratio P4 secretion/cell viability of 0.83 at 2.5 µg/mL (0.4 μg/cm^2^), 1.19 at 10 µg/mL (1.6 μg/cm^2^), 1.14 at 40 µg/mL (6.3 μg/cm^2^) and 2.32 at 320 µg/mL (50.4 μg/cm^2^)). This was not the case for E2, which also followed the profile of cell viability, with a decreased secretion mainly at the concentration of NP of 320 µg/mL (50.4 μg/cm^2^). We next evaluated the protein expression of all the enzymes involved in the pathways of steroidogenesis (Figure 7B). Intracellular cholesterol is carried into the mitochondria by metastatic lymph node 64 protein (MLN64, also known as StAR-related lipid transfer domain containing 3 (STARD3)), where it is cleaved by cytochrome P450 side-chain cleavage (P450ssc) to form pregnenolone, which is converted to P4 by 3β-hydroxysteroid dehydrogenase (HSD3β1). E2 is produced from dehydroepiandrosterone sulfate (SDHEA) by the steroid sulfatase (STS) into DHEA and then by two last enzymes 17β-hydroxysteroid dehydrogenase 1 (HSD17β1) and aromatase (CYP19A1). Western blot analyses show that all of these enzymes are present in both VCT and ST, but their protein levels are increased during ST formation, while MLN64 does not change (Figure 7B), as already described [37]. Figure 7B shows a significant decrease in aromatase protein levels from 40 µg/mL (6.3 μg/cm^2^) (0.82 ± 0.03 and 0.43 ± 0.02 AU with 320 µg/mL (50.4 μg/cm^2^), respectively). Again, a decrease in the protein levels of HSD3β1 (0.80 ± 0.06 AU), HSD17β1 (0.73 ± 0.09 AU) and STS (0.64 ± 0.12 AU) was obtained with the highest concentration tested (320 μg/mL (50.4 μg/cm^2^)), without returning to the initial levels of the undifferentiated VCT, except for STS. Gene expressions of aromatase and P450scc were also evaluated by qPCR (Appendix A) and no significant variations were observed with the concentrations between 2.5 to 40 µg/mL (0.4 to 6.3 μg/cm^2^), in accordance with the WB results. Altogether these results show that CeO_2_ NPs decrease hCG, hPL, P4 and E2 secretion only at high and toxic concentrations, but disturb hCG and hPL at much lower concentrations (40 µg/mL (6.3 µg/cm^2^)), for which the metabolic activity of the trophoblast was affected in the absence of LDH release.

## 4. Discussion

In the current study, we investigated the impact of cerium dioxide nanoparticles on human placenta by using human primary trophoblasts isolated from placentas obtained after cesarean section at term. Our study is the first to use a physiological model to investigate the effect of NP, and in particular of CeO_2_ NPs, on the human placental barrier. The use of primary cells to assess the toxicity of NPs is of particular interest because these cells have not undergone any modification and better represent the in vivo state. Cytotrophoblasts isolated from placenta at term of pregnancy have the spontaneous capacity to aggregate and fuse in vitro after 24–48 h of culture to form the syncytiotrophoblast, and are thus more physiological than the BeWo trophoblast cell line, which requires the addition of forskolin, an activator of PKA, in order to fuse. Moreover, the primary villous cytotrophoblasts are non-proliferative, and exit from the cell cycle in an intermediary differentiated state, which affects the cellular responses in comparison to the proliferative trophoblasts cell lines. These cell lines are choriocarcinoma and usually more resistant to apoptosis induction and less close to physiologic reality.

A growing body of epidemiological evidence has established a causal relationship between the mother’s exposure to air pollution (with concentrations of particulate matter (PM_2.5_) that range from 1.8 to 22.1 μg/m^3^) and the increased risk of adverse pregnancy outcomes [38,39], such as preterm birth, intrauterine growth retardation (IUGR), mortality of infants and/or fetuses. NPs are able to cross biological barriers (pulmonary and intestinal) [40], enter the systemic circulation and reach secondary organs, such as the placenta. For instance, NPs administered via different routes (e.g., intravenous injection, intratracheal instillation or oral administration) in rodents can circulate in the blood stream and reach other organs [41,42,43]. The placenta acts then as a second barrier for the fetus against these environmental substances. But a number of studies from animal models have already demonstrated that some NP can cross the placenta [19,44] and cause fetotoxicity [45]. However, even in the absence of passing through the placental barrier, pollutants that accumulate within the placenta can directly impact the proper functioning of this organ, and placenta dysfunction has harmful implications for fetal development and a major role in the pregnancy outcome [46]. Moreover, it is now admitted that the early life environment can also impact the risk of developing chronic diseases from childhood to adulthood, in the concept termed DOHaD (for Developmental Origins of Health and Disease). Thus, exposure to environmental chemicals, especially the novel NPs, is of particular interest for this sensitive period of pregnancy.

Today, the novel CeO_2_ NPs are found in our everyday lives in several consumer products [7], but pregnant women are mainly exposed to CeO_2_ from air pollution [47,48]. Thus, most of the studies on CeO_2_ NP toxicity have focused on cellular models like the lung [49,50,51,52,53], and their effects during the narrow period of human pregnancy, and particularly on the human placenta, are poorly understood. Cerium (Ce) has been detected in blood and urine samples and breast milk of pregnant women, suggesting that the cerium could be also present in the human placenta, although it has not yet been evaluated there [14,15,16]. Currently, there are no data on the presence of metallic nanoparticles, including Ce, in the human placenta, but a recent study has already pointed out the presence of carbon black particles in the placentas of women exposed to particulate air pollution, suggesting the translocation of ultrafine particles from the mothers’ lungs into the circulation and then to the placenta [54]. Moreover, a recent in vivo study provides the first evidence that respiratory exposure to metallic nanoparticles (e.g., CeO_2_, TiO_2_, and Ag) during pregnancy in mice leads to accumulation of all of these NPs in the placenta and to long-lasting lung impairment in the offspring [18]. Moreover, recent research indicates that maternal exposure to CeO_2_ NPs during early pregnancy induces placental abnormalities and impairs the pregnancy outcome in mice [17]. However, placental structure morphologically and functionally differs highly among species and data from animal models cannot always be extrapolated to humans, and human placental models are needed to explore the mechanisms. For instance, the secretion of steroids, hCG and hPL is specific to human placenta, but in rodents steroids are produced by the ovary all along the gestation and there is no placenta relay.

In the present study, CeO_2_ NPs were used at a large concentration range from 0.6 to 640 μg/mL, corresponding to 0.1–101 μg/cm^2^ (Table 1). The lowest values were chosen to represent a realistic situation in accordance with the data available in the literature. For example, the retention of carbon NPs in the lung is up to 50 μg/cm^2^ after chronic exposure [55] and the transfer rate to the blood stream was estimated to be 0.1% [56], thus the amount that could reach the placenta is around 0.05 μg/cm^2^. However, the in vivo biokinetic study (rat model) showed a half-life of 22 days for metallic NPs (ferric oxide) in the blood stream [57]. Therefore NPs could accumulate in the blood and higher levels could be reached over time in the placenta [58]. We also used higher concentrations in this study that allowed us to make a complete cytotoxicity evaluation to determine the IC50 of CeO_2_ NPs, which was not reached at 24 h of exposure and was 320 μg/mL (50.4 μg/cm^2^) at 48 h. These high concentrations also allow further comparison of toxicity between placental and other barriers (especially pulmonary or intestinal) and may correspond to a worst-case scenario knowing that CeO_2_ NPs are now being proposed as potential medicine (with actually-tested concentrations reaching up to 250 µg/mL) [59].

Particle size and surface charge are important factors accountable for a range of biological effects of NPs, like cellular uptake and toxicity [60,61]. Thus, physicochemical characterization of the CeO_2_ NPs used in our study was done prior to the cell culture experiments. Our results showed that NPs form agglomerates around 400–500 nm in all tested media (directly after sonication) (Figure 1), which are significantly larger than the individual NPs, which are about 30 nm in size (Table 3). The hydrodynamic size remains substantially identical after 1 h, indicating stability of the formed agglomerates at least in the short term. In addition, the size measured by DLS showed that the trophoblasts are mostly in contact with agglomerates of NPs. Moreover, the high surface area of NPs is important to consider because it allows adsorption of proteins, forming a corona, as already described [62,63], which plays a key role in bioprocessing of NPs: dissolution in media, uptake and biodistribution and toxicity [64]. CeO_2_ NPs have a negative surface charge in solution, as determined by zeta potential measurements (Table 3), but only of around -30 mV, which is not sufficient to prevent the agglomeration of particles by repulsive forces. It has been established that the surface charge can influence the internalization of NPs and, for instance, the negative charge of TiO_2_ NP has been shown to prevent their uptake by epithelial cells [65]. It was proposed that aggregation could prevent the NPs from binding to the cell surface and entering the cell, thus reducing toxicity [66]. However, when exploring the internalization capacity of NPs by both villous mononucleated cytotrophoblasts and the differentiated multinucleated syncytiotrophoblast, we showed that these cells were able to internalize NPs, no matter what the concentration of NPs tested (0.6, 2.5, 10 and 40 µg/mL, corresponding to 0.1, 0.4, 1.6 and 6.3 μg/cm^2^, respectively), harboring on average 2–3 agglomerates of NPs per cell, found only in the cytoplasm in a perinuclear zone (Figure 2 and Figure 3). No NPs were, however, detected in the nucleus despite this perinuclear localization, and we could not detect NP in the mitochondria or endoplasmic reticulum. We then measured the intracellular NP agglomerates and found a wide distribution, depending on the concentration of NPs, ranging from 10^2^ to 10^6^ nm^2^ corresponding to a size of 10 to 1000 nm (Figure 2D). We also observed that NPs formed larger agglomerates in cells than in extracellular media (DLS measurements), suggesting that NPs accumulate within the cells. However, the majority of NPs were found individually but grouped in agglomerates without forming a compact mass (surface area ranging from 140 to 10^3^ nm^2^, corresponding to the primary size of NPs) (Figure 2 and Table 1). We confirmed after chemical analysis and elemental mapping by STEM and EDX spectroscopy that the agglomerates visible in the cells did indeed comprise CeO_2_ NPs (Figure 3).

As we have demonstrated that CeO_2_ NPs are taken up by human trophoblasts (0.6 to 40 μg/mL (0.1 to 6.3 μg/cm^2^)), we explored CeO_2_ NPs’ toxicity on these cells using two complementary tests, a metabolic activity (WST1) test and an evaluation of the membrane integrity (LDH release). We found that the cytotoxicity of CeO_2_ NPs in primary trophoblasts is time- and concentration-dependent (Figure 4A). For short time exposure (24 h), cytotoxicity was observed only for concentrations usually tested in pulmonary models (80–640 μg/mL (12.6–101 μg/cm^2^)), a tissue in direct contact with inhaled air. It is unlikely that the placenta could be exposed to such high NP concentrations from air sources, although some high concentrations of CeO_2_ could be reached in medical treatments (e.g., antitumoral drugs). For longer periods of exposure (48 h), however, lower NP concentrations (from 5 µg/mL (0.8 μg/cm^2^)) affect the trophoblast metabolic activity (WST1 test), suggesting that longer exposure to low concentrations could impair placental integrity and functions. Our primary trophoblasts culture model does not allow us to make longer exposure times (longer than 72 h), to investigate the effects of chronic exposure of the placenta to CeO_2_ NPs. We have not found any other reports of the toxicity of CeO_2_ NPs on placenta. Therefore, we will compare our observations to the effects of other metallic oxides’ NPs. TiO_2_ NP did not change cell viability at 10 and 100 µg/mL at both 24 h and 48 h in human trophoblast HTR-8/SVneo cells [20], while iron oxide NP decreased cell viability from 3 µg/cm^2^ (which would correspond to 20 µg/mL in our study) at 24 h of exposure in the trophoblast cell line BeWo [67]. Molecular pathways activated by NPs are mainly necrosis, apoptosis or autophagy depending on the concentration and the chemical nature of the NP tested [68,69]. In our study, the toxicity of CeO_2_ NPs on trophoblasts involved the activation of caspases for NP concentrations >10 µg/mL (Figure 4C,D), suggesting a commitment to apoptosis. The absence of p53 stabilization at most of the concentrations tested (10, 40 and 320 µg/mL (1.6, 6.3 and 50.4 μg/cm^2^, respectively)) (Figure 4D) suggests that apoptosis occurs in a p53-independent pathway and, thus, apparently without genotoxicity.

The two oxidation states of cerium (Ce^3+^ or Ce^4+^) enable the formation of oxygen vacancies, which are essential to its high reactivity and its ability to act as a catalyst for both oxidation and reduction reactions [70]. However, excessive ROS generation can lead to oxidative stress by altering the fine balance between pro- and anti-oxidant species and can cause cell death. Normal pregnancy is characterized by low-grade oxidative stress and sustained oxidative stress is recognized as a key factor in the pathogenesis of many pregnancy-related disorders such as intrauterine growth restriction (IUGR), pre-eclampsia and miscarriage [71]. The effects of CeO_2_ NPs on oxidative stress are still controversial since both prooxidant [72,73] and antioxidant [50,74] effects have been reported, depending on the tissue models employed and NP physico-chemical characteristics. In this article, we show that CeO_2_ NPs (representative test material NM-212) are antioxidant in human primary trophoblasts, especially at high concentrations and after an incubation period of 24 h to 72 h (Appendix A). Surprisingly, for a shorter exposure time (4 h) and at low concentrations (5–20 µg/mL (0.8–3.2 μg/cm^2^)), NP seems to be pro-oxidant (Figure 5), but this effect is temporary, because it is no longer visible at longer exposure times (from 24 h). We then depicted the antioxidant system defenses by following the protein levels of the main enzymes, such as SOD1, SOD2 and catalase. We also explored HO-1, which is induced in response to oxidative stress, and Prx-SO_3_, an oxidative stress sensor. We observed no changes in protein levels of these enzymes and sensors (Appendix A). Therefore, CeO_2_ NPs do not appear to induce oxidative stress in trophoblasts when applied at low concentrations and act instead as ROS scavengers at high concentrations. Indeed, the redox balance is essential for many biological processes. Oxidative stress has been related to several disorders leading to placental dysfunctions, but recent papers suggest that an imbalance in favor of reductive stress could also contribute to deleterious effects (e.g., in cardiomyopathies) [75,76]. 

The integrity and renewal of the human placental barrier during pregnancy involves the fusion of the underlying VCT with the existing ST. In culture, VCT spontaneously aggregate and then fuse to form nonproliferative, multinucleated and hormonally active ST [31]. In this paper, we show a decrease of approximately 20% in the ability of VCT to fuse after in vitro exposure to CeO_2_ NPs, whatever the concentration tested (2.5 and 40 µg/mL (0.4 and 6.3 μg/cm^2^)) (Figure 6). Thus, CeO_2_ NPs that impact trophoblast fusion and affect VCT differentiation in ST could compromise the successful renewal of the syncytium all along the pregnancy, and lead to early aging of the placenta. The effect on the trophoblast differentiation/fusion has repercussions, especially at the beginning of pregnancy, because any impact on the formation and renewal of the syncytium in the first trimester could lead to placental abnormalities and long-term functional consequences. Moreover, disruption of the differentiation capacity of the VCT to renew the ST by a cell fusion process, or impairment of the endocrine functions exerted by the ST, have been associated with pregnancy disorders like preeclampsia (PE) or intrauterine growth restriction (IUGR) [77,78]. Some mechanisms have been reported to explain trophoblast fusion, involving in particular the syncytins 1 and 2 proteins, with fusogenic properties, and their receptors (ASCT2 and MFSD2, respectively). The effect of CeO_2_ NPs on trophoblast fusion does not seem to deregulate expression of syncytins 1/2 and their receptors, at their transcript levels (Appendix A).

The gene expression of two other factors involved in the process of trophoblast differentiation has also been evaluated: GCM1 and CDH1. GCM1 has a central role in mediating the differentiation of trophoblast cells along both the villous and extravillous pathways in placental development, with its action as a transcription factor to control the expression of placenta-specific genes, in particular PlGF [79]. The CDH1 gene, which encodes cadherin-1 protein, is involved in mechanisms regulating cell–cell adhesion, mobility and proliferation of epithelial cells [80]. No change in gene expression of these two factors after trophoblast incubation with CeO_2_ NPs was observed. Trophoblast fusion also involves proteins of adherent junctions like E-cadherin, tight junctions like zona occludens (ZO-1) or gap junctions like connexin-43 [81,82]. Thus, it would be interesting to follow the levels and localization of these proteins involved in the intercellular junctions after NP exposure. However, we can also suppose that heavy aggregates of NPs could mechanically prevent the trophoblast from moving, aggregating and fusing, leading to a decreased fusion index in culture.

We investigated the endocrine function of the trophoblast at three levels: peptide hormones (with hCG and hPL), angiogenic factor (with PlGF) and steroid hormones (with P4 and E2) (Figure 7). The decrease in VCT fusion observed for concentrations of 40 μg/mL (6.3 μg/cm^2^) disturb the endocrine function, for hCG and hPL, the latter only produced by the ST, with a diminution of around 20% of their secretion, but not of the other hormones tested. For the concentration of 320 μg/mL (50.4 μg/cm^2^), hormonal secretion followed the decrease of the metabolic activity of trophoblasts, except for P4, whose secretion was maintained, suggesting a compensatory adaptation in the production of this steroid. Hormonal disturbances are associated with some fetal complications of pregnancy, i.e., the level of hCG is decreased in IUGR while it is increased in pre-eclampsia [83]. hCG is the first placental hormone secreted by the trophoblast and plays a pleiotropic role during pregnancy (e.g., a role in the maintenance of P4 production by the corpus luteum in early pregnancy, modulation of uterine immune responses) [84]. Moreover, hCG plays an essential role in the differentiation of the trophoblast, by an autocrine mechanism, and promotes further differentiation of cytotrophoblasts into syncytiotrophoblasts [85]. P4 allows maintenance of myometrial quiescence and E2 regulates utero-placental blood flow and plays a role in contraction of the myometrium during childbirth [86]. Therefore, disturbances of these hormones, especially in early pregnancy, can affect the course of pregnancy and its outcome. For hCG, there were high inter-individual placenta variations (0.4 to 1.2 AU) at concentrations of NP below to 10 μg/mL (1.6 μg/cm^2^), although the average of the results from nine placentas were insignificant, which could not be explained by the placental sex differences in their response to NPs. Even though we showed that CeO_2_ NPs induce a decrease in hCG secretion in placentas from terminal pregnancy, it would be even more interesting to assess the effects of CeO_2_ NP on placentas from first trimester of pregnancy.

Steroid hormones, initiated in the mitochondria (for P4) or produced in the cytosol (E2), are immediately released into the extracellular medium after their production. This process is faster than hCG production because it does not require transcription, translation and glycosylation processes. This could explain the maintenance of the secretion of P4 at the cytotoxic concentration of 320 μg/mL (50.4 μg/cm^2^) (Figure 7A), unlike the observed decrease for hCG secretion at the same concentration of NPs. The maintenance of secretion of P4 is correlated with the fact that there is no major variation in the protein expression of the enzymes required for its production (MLN64 and HSD3β1), although there is for those involved in E2 production (STS, HSD17β1 and aromatase), in agreement with the decreased E2 secretion (Figure 7A,B). This adaptive process for P4 may involve the mitochondria, as the main site of steroidogenesis. This perturbation of the main hormones of pregnancy, such as hPL and hCG with the concentrations of NPs (40 μg/mL (6.3 μg/cm^2^)), and the decrease in hCG, P4, E2 and PlGF hormones with much higher concentrations of NPs (320 µg/mL (50.4 μg/cm^2^)), suggests that CeO_2_ NPs could impair the endocrine function after perturbed metabolic activity or due to NP toxicity. Our study also investigated some mechanisms of CeO_2_ NPs’ toxicity like caspase activation, ROS production and followed the main anti-oxidative enzymes. Moreover, our investigations on the actors of trophoblast fusion (syncytins in particular) and all of the enzymes of steroidogenesis also failed to reveal the mechanisms of action of CeO_2_ NPs in human trophoblasts.

## 5. Conclusions

This study is the first to address the toxicity of CeO_2_ NPs on human primary trophoblasts, the mechanisms of their toxicity (p53 and caspase activation), their effect on oxidative stress (ROS production and the main antioxidant enzymes) and their impact on placental barrier functions in its globality (trophoblast differentiation capacity, hormone secretion), together with the main genes and enzymes that assure the differentiation and endocrine function of ST. Here, using both TEM and STEM/EDX imaging, we demonstrate the ability of both cytotrophoblasts and syncytiotrophoblasts to internalize cerium dioxide nanoparticles in a concentration-dependent manner. CeO_2_ NPs induce a time- and concentration-dependent toxicity on primary human trophoblasts, by caspases activation and in the absence of oxidative stress. Instead, CeO_2_ acts as a ROS-scavenger at high concentrations. Moreover, we showed that CeO_2_ NP perturbed trophoblast syncytialization and secretion of the pregnancy hormones such as hCG, hPL, PlGF P4, and E2, which altogether can lead to trophoblast dysfunction and poor prognosis in the pregnancy health outcome. With the relationships demonstrated between exposure to environmental pollution and risks for adverse pregnancy outcomes, we need to understand all aspects, such as the uptake, toxicity, mechanism of action and impact on physiologic activity in order to comprehend the biological effects of NPs on the human placenta. All these data help to assess the risk of exposure to CeO_2_ NPs in pregnant women.

## Figures and Tables

**Figure 1 nanomaterials-10-01309-f001:**
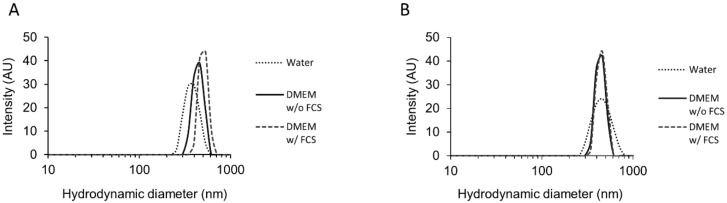
Size distributions of cerium dioxide nanoparticles in solution. Nanoparticles were suspended in cell culture media without fetal calf serum (FCS) (DMEM w/o FCS, 3 mg/mL), sonicated and diluted (0.05 mg/mL) in water, DMEM w/o FCS or cell culture media with FCS (DMEM w/FCS). Dynamic light scattering (DLS) analysis of cerium dioxide nanoparticles (CeO_2_ NPs) showing size distributions and hydrodynamic diameters of NPs immediately after sonication (**A**) or 1 h after sonication (**B**).

**Figure 2 nanomaterials-10-01309-f002:**
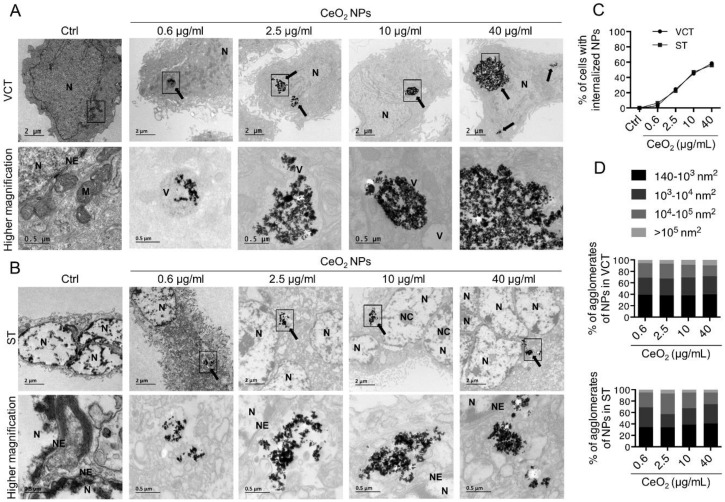
Cerium dioxide nanoparticles uptake by human primary cytotrophoblast and syncytiotrophoblasts. Primary cytotrophoblasts (VCT) purified from term placenta were plated overnight and cultured for 24 h to obtain VCT or for 72 h in order to form the syncytiotrophoblast (ST). VCT or ST were either untreated (Ctrl) or incubated with cerium dioxide nanoparticles (CeO_2_ NPs) at the indicated concentrations for 24 h. (**A**) Transmission electron microscopy (TEM) photomicrographs, with applied auto-contrast, of VCT treated as described. Bottom images are higher magnifications of the squares; images are rotated by around 90° to the right. Black arrows show the agglomerates of nanoparticles. Scale bars: 2 μm and 500 nm. (**B**) TEM photomicrographs, with applied auto-contrast, of ST treated as described. Bottom images are higher magnifications of the squares; images are rotated by around 90° to the right. Black arrows show the agglomerates of nanoparticles. Scale bars: 2 μm and 500 nm. (**C**) Percentage of VCT and ST with internalized nanoparticles from quantification of 200 cells under TEM. (**D**) Measurement of the size of nanoparticles and/or agglomerates in VCT and ST, represented as percentage of occurrence according to the surface area. M: mitochondrion; N: nucleus; NC: nucleolus; NE: nuclear envelope; V: vesicle.

**Figure 3 nanomaterials-10-01309-f003:**
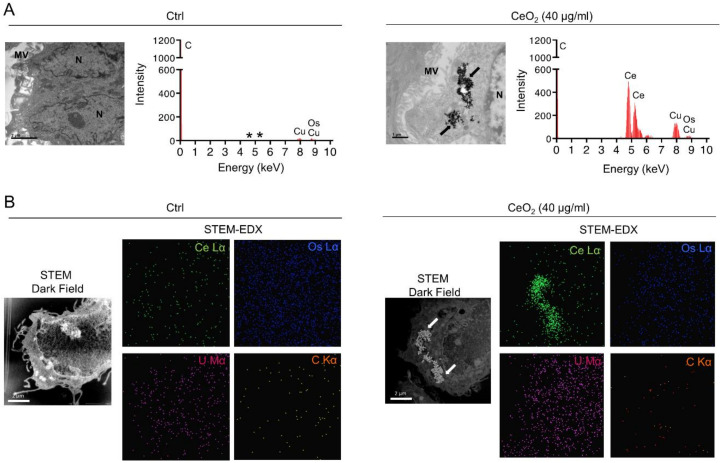
EDX and STEM analysis of cerium dioxide nanoparticles inside cells. VCT purified from term placenta were plated overnight in culture and then were either untreated (Ctrl, left) or incubated with 40 μg/mL of CeO_2_ NPs (right) for 24 h. (**A**) Transmission electron microscopy (TEM) photomicrographs and graphs of the corresponding EDX spectra. The absorption peaks are labeled with symbols of the element detected: Carbon (C), Cerium (Ce), Copper (Cu), Osmium (Os). Stainings corresponding to Os, U and C serve as control. Black arrows show the agglomerates of nanoparticles in TEM. Scale bars: 2 μm and 1 μm. (*) mark the expected cerium energy peaks. (**B**) Images of Scanning Transmission Electronic Microscopy (STEM) analysis under darkfield z-contrast and the corresponding EDX mapping for cerium (Ce Lα), osmium (Os Lα), uranium (U Mα) and carbon (C Kα) in the scanned area. White arrows show the agglomerates of nanoparticles in STEM. Scale bars: 2 μm. MV: microvilli; N: nucleus.

**Figure 4 nanomaterials-10-01309-f004:**
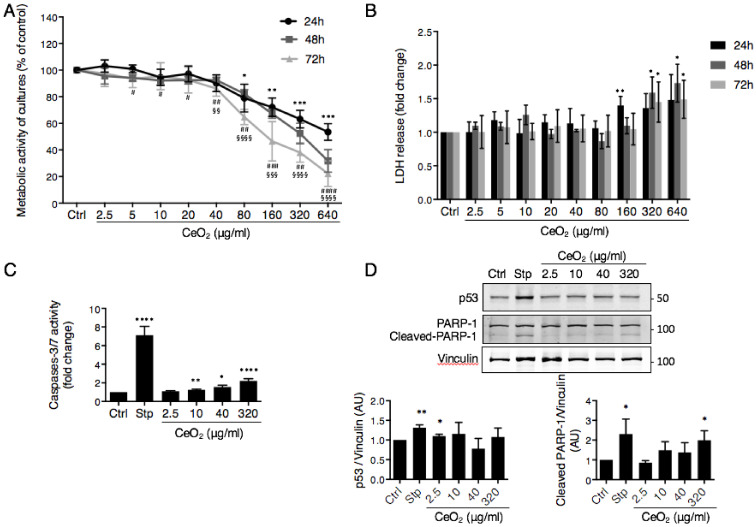
Effect of cerium dioxide nanoparticles on human primary trophoblast viability. VCT purified from term placenta were plated overnight in culture and were either untreated (Ctrl) or incubated with CeO_2_ NPs at the indicated concentrations for 24 h to 72 h. (**A**) WST1 assay showing metabolic activity of VCT as means +/− SEM (n = 3) relative to control for each time of treatment. * *p* <0.05; ** *p* < 0.01; *** *p* < 0.001 vs 24 h; # *p* <0.05; ## *p* <0.01; ### *p* < 0.001; #### *p* < 0.0001 vs 48h; §§ *p* < 0.01; §§§ *p* < 0.001; §§§§ *p* < 0.0001 vs 72 h. (**B**) LDH release in culture media showing VCT membrane permeabilization after exposure to nanoparticles. Data are represented as means +/− SEM (n = 4) relative to untreated cells (Ctrl). * *p* < 0.05; ** *p* <0.01 vs Control. (**C**) Caspase-3/7 activity of VCT evaluated by ApoONE assay. Staurosporine (Stp, 0.5 μM, 4 h of incubation) was used as positive control. Fluorescence emission was recorded and the graph represents the mean +/− SEM (n = 4) normalized to untreated cells (Ctrl). (**D**) VCT were left untreated (Ctrl), were incubated with CeO_2_ NPs at the indicated concentrations for 24 h or with staurosporine (Stp, 0.5 μM) for 4 h. Total protein extracts were subjected to SDS-PAGE under reducing conditions and membranes were immunoblotted with anti-p53, anti-PARP-1 and anti-vinculin antibodies (the latter used as loading control). Immunoblots were quantified with an Odyssey System Imager and are shown in the bar scale graph as ratio to the vinculin as mean +/− SEM (n = 3). * *p* < 0.05; ** *p* < 0.01 vs. Control.

**Figure 5 nanomaterials-10-01309-f005:**
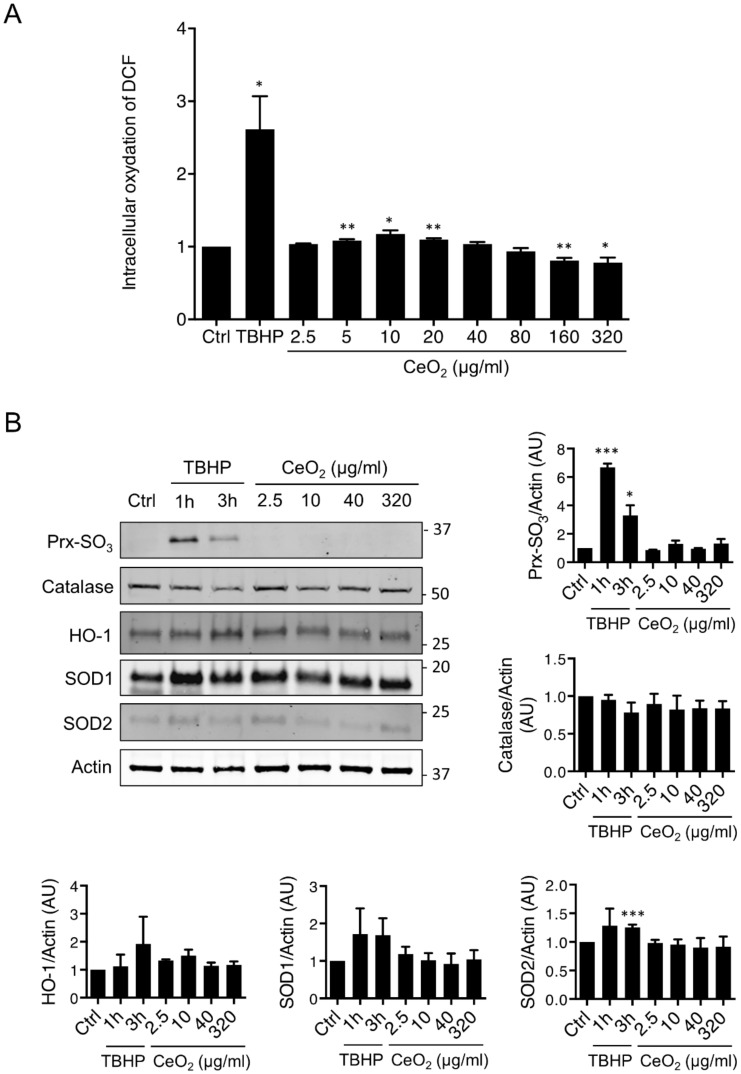
Intracellular oxidative stress after trophoblast exposure to cerium dioxide nanoparticles. VCT purified from term placenta were plated overnight in culture and then left untreated (Ctrl) or incubated with CeO_2_ NPs at the indicated concentrations for 4 h. (**A**) Intracellular ROS levels of VCT were evaluated using CM-H_2_DCFDA. Tert-butyl hydroperoxide (TBHP, 250 μM) was used as positive control. Experiments were performed in technical triplicate from 3 independent trophoblast cultures (n = 3). Data are expressed as means +/− SEM normalized to control. * *p* < 0.05; ** *p* < 0.01. (**B**) VCT were untreated (Ctrl), treated with CeO_2_ NPs at the indicated concentrations for 4 h or with tert-Butyl hydroperoxide (TBHP, 100 μM) for 1 h or 3 h of incubation. Total protein extracts were subjected to SDS-PAGE under reducing conditions and membranes were immunoblotted with anti-Prx-SO_3_, anti-catalase, anti-HO1, anti-SOD1, anti-SOD2 and anti-actin antibodies (the latter used as loading control). Immunoblots were quantified with an Odyssey System Imager and are shown in the bar scale graph as the ratio to actin as mean +/− SEM (n = 3).

**Figure 6 nanomaterials-10-01309-f006:**
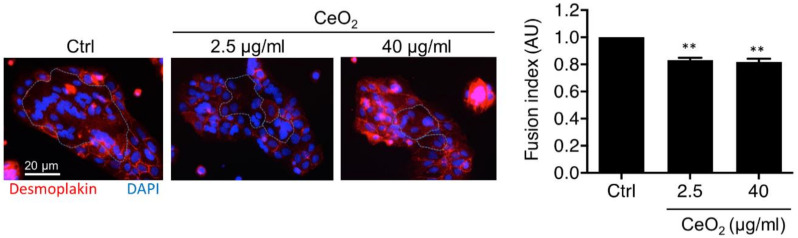
Impact of cerium dioxide nanoparticles on trophoblast fusion. VCT purified from term placenta were plated overnight in culture and were either untreated (Ctrl) or incubated with CeO_2_ NPs at the indicated concentrations for 72 h. ST were then fixed and permeabilized with methanol. Epifluorescence microscopy images after desmoplakin immunostaining (membrane protein, red) and nuclei (DAPI, blue). The dotted lines delimit the syncytiotrophoblast. Scale bar: 20 μm. Nuclei were counted using ImageJ. Fusion index was calculated and values are represented as mean +/− SEM (n = 3) of control (right graph). ** *p* < 0.01.

**Figure 7 nanomaterials-10-01309-f007:**
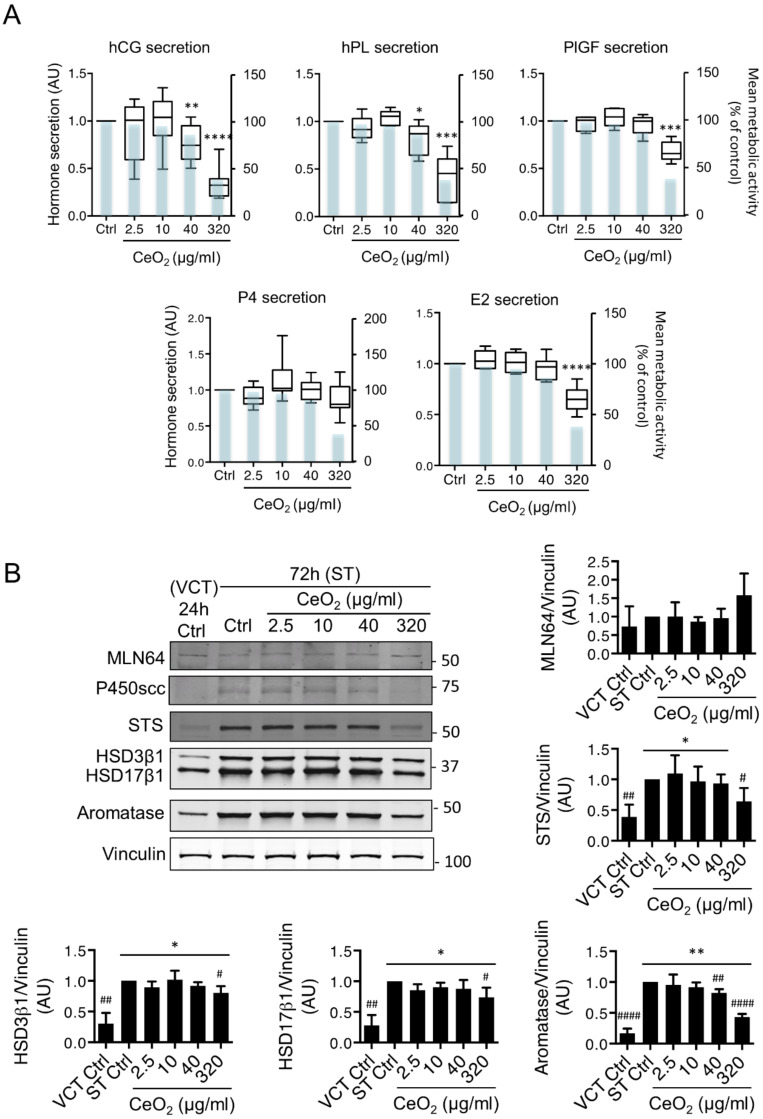
Effect of cerium dioxide nanoparticles on trophoblast endocrine function. VCT purified from term placenta were plated overnight in culture and then were either untreated (Ctrl) or incubated with CeO_2_ NPs at the indicated concentrations for 72 h. (**A**) After 72 h of culture, dosage of human chorionic gonadotropin (hCG), human placental lactogen (hPL), placenta growth factor (PlGF), progesterone (P4) and estradiol (E2) were assayed in ST supernatants. Results are represented in the boxplots as medians and lower and upper quartile +/− minimum and maximum (n = 9 for hCG and P4, n = 6 for E2, n = 5 for PlGF and hPL) with respect to control. ** *p* < 0.01; *** *p* < 0.001; **** *p* <0.0001 as compared to controls. The overlapping graphs represent the corresponding average values of trophoblast metabolic activity as determined by WST1 and shown in Figure 4A. (**B**) Total protein extracts from trophoblasts were subjected to SDS-PAGE under reducing conditions and membranes were immunoblotted with anti-MLN64, anti-P450scc, anti-HSD3β1, anti-HSD17β1, anti-aromatase and anti-vinculin (as loading control) antibodies. Immunoblots with normalization to control at 72 h were quantified using an Odyssey System Imager and shown in the bar scale graphs as the ratio to vinculin as mean +/− SEM (n = 3). * *p* < 0.05 and ** *p* < 0.01 vs. VCT Ctrl; ^#^
*p* < 0.05; ^##^
*p* < 0.01; ^###^
*p* < 0.001 and ^####^
*p* < 0.0001 vs ST Ctrl.

**Table 1 nanomaterials-10-01309-t001:** Nanoparticle (NP) concentrations used in μg/mL and correspondence in μg/cm^2^.

CeO_2_ (μg/mL)	Corresponding CeO_2_ Concentrations in μg/cm^2^
0.6	0.1
2.5	0.4
5	0.8
10	1.6
20	3.2
40	6.3
80	12.6
160	25.2
320	50.4
640	101

**Table 2 nanomaterials-10-01309-t002:** Primer sequences.

ASCT2	forward 5′-GGCTTGGTAGTGTTTGCCAT-3′
	reverse 5′-GGGCAAAGAGTAAACCCACA-3′
Aromatase	forward 5′-CCTGAAGCCATGCCTGCTGC-3′
	reverse 5′-CCGATCCCCATCCACAGGAATCT-3′
CDH1	forward 5′-CATTGCCACATACACTCTCTTCT-3′
	reverse 5′-CGGTTACCGTGATCAAAATCTC-3′
GCM1	forward 5′-CAGAAGCAGCAGCGGAAAC-3′
	reverse 5′-GACCTCGGCAAGGGATGA-3′
hCG beta	forward 5′-GCTACTGCCCCACCATGACC-3′
	reverse 5′-ATGGACTCGAAGCGCACATC-3′
hPL	forward 5′-GCATGACTCCCAGACCTCCTT-3′
	reverse 5′-TGCGGAGCAGCTCTAGATTGG-3′
HPRT	forward 5′-TGACACTGGCAAAACAATGCA-3′
	reverse 5′-GGTCCTTTTCACCAGCAAGCT-3′
MFSD2	forward 5′-CTCCTGGCCATCATGCTCTC-3′
	reverse 5′-GGCCACCAAGATGAGAAA-3′
P450scc	forward 5′-TTTTTGCCCCTGTTGGATGCA-3′
	reverse 5′-CCCTGGCGCTCCCCAAAAAT-3′
PlGF	forward 5′-GCTCGTCAGAGGTGGAAGTGGT-3′
	reverse 5′-CTCGCTGGGGTACTCGGACA-3′
RPL13	forward 5′-AAGGTCGTGCGTCTGAAG-3′
	reverse 5′- GAGTCCGTGGGTCTTGAG-3′
RPLO	forward 5′-AACATCTCCCCCTTCTCCT-3′
	reverse 5′-ACTCGTTTGTACCCGTTGAT-3′
sFlt1	forward 5′-ACAATCAGAGGTGAGCACTGCAA-3′
	reverse 5′-TCCGAGCCTGAAAGTTAGCAA-3′
SDHA	forward 5′-TGGGAACAAGAGGGCATCTG-3′
	reverse 5′-CCACCACTGCATCAAATTCATG-3′
Syncytin 1	forward 5′-CGGACATCCAAAGTGATACATCCT-3′
	reverse 5′-TGATGTATCCAAGACTCCACTCCA-3′
Syncytin 2	forward 5′-GCCTGCAAATAGTCTTCTTT-3′
	reverse 5′-ATAGGGGCTATTCCCATTAG-3′

**Table 3 nanomaterials-10-01309-t003:** Physicochemical characteristics of cerium dioxide nanoparticles.

			Zeta Potential (mV)			Hydrodynamic Diameter (nm)			
Primary Particle Size (ø in nm) ^a^	BET Surface Area (m^2^/g) ^a^	Crystalline Structure ^a^	Water	DMEM w/o FCS	DMEM w/FCS	Water		DMEM w/o FCS	DMEM w/FCS
28.4 ± 10.4	27.2 ± 0.9	polyhedral	−32.7 ± 0.9	−21.7 ± 2.8	23.4 ± 1	Time 0: Time 1h:	378 ± 65 441 ± 43	441 ± 57 411 ± 51	503 ± 55 443 ± 80
						Time 24 h:	804 ± 30	aggregation	aggregation
						Time 48 h:	758 ± 47	aggregation	aggregation
						Time 72 h:	736 ± 29	aggregation	aggregation

^a^ Manufacturer’s data; BET: Brunauer, Emmett and Teller method; w/: with; w/o: without.

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
