# Peer review of "Uptake of Cerium Dioxide Nanoparticles and Impact on Viability, Differentiation and Functions of Primary Trophoblast Cells from Human Placenta"

_nanomaterials, 2020, doi:10.3390/nano10071309_

Round 1

Reviewer 1 Report

The manuscript be Nedder et al investigates the potential effects of Cerium Dioxide Nanoparticles on primary human trophoblast cells. The authors characterize the NPs, its incorporation in trophoblast cells primary cultures and its effects on several parameters including proliferation, caspase activation, oxidative stress markers and endocrine function, among others.

The manuscript is well written, several determinations on the effects of Cerium NPs on trophoblast cells are performed. The conclusions are supported by the results. The authors conclude the cerium NPs are toxic in a time- and concentration-dependent fashion on primary human trophoblasts, mechanisms proposed involved caspases activation and absence of oxidative stress.

This data set adds to the important research on the effects of environment pollutants on placental function and potential pregnancy outcomes.

Some text mistakes were observed, please see below for additional format suggestions.

Line 360, it’s referred as Table 1 containing information on physicochemical data, however Table 1 only shows the equivalence between concentration and area covered by NPs considering the surface of the wells used. The authors probably are referring to Table 3.

Lines 397 and 472, in this case it is indicated as Table 2 when referring to the equivalence between NPs concentration and surface. It should be Table 1.

Line 529 Figure 2A probably refers to Figure S2A.

Line 702 Delete “In this project…”

Line 832 Delete “…”

Line 857 Delete “deep”

Line 719 the section “…which are about 30 nm in size (Table 1)” probably refers to Table 3.

Line 726 the section “…as determined by zeta potential measurements (Table 1)” probably refers to Table 3.

Line 859 Delete "..far.."

The level of resolution in some figures needs to be improved, for example in Figure 7. 

Discussion should start with the main findings of this study. The authors only begin to mention the main findings in the third paragraph of discussion.

Reviewer 2 Report

This is an interesting study describing the Uptake of Cerium Dioxide nanoparticles and impact on viability, differentiation and functions of primary trophoblast cells from human placenta.

However bevor it can be published some Points should be revised:

The Figure legends of Figures 2-7 are not self explaining. Some Major issues are missing:

Figure 2: Figure 2 is divided in A, B, C, D. What is shown in the 4 different parts. The figures contain arrows and boxes, what is shown there? Some Figures contain alphabetic characters (N, V, NC, NE), what is the reason? What is the meaning of VCT and ST? Please explain and please do not copy the text from the results chapter!

Figure 3: Figure 3 is also divided in A and B without Explanation in the legend! Please revise. In Addition, what is shown by the White arrow? In B 8 different fluorescence stainings are shown but 7 appear to be without any staining. What is the reason for showing empty stainings?

Figure 4: Figure 4 is also divided in A, B, C, D. What is shown in the 4 different parts. Please explain! 

Figure 5: Briefly explain the different parts of Figure 5 in the legend. Figure 5 has a very low resolution could you please enhance the font size!

Figure 6: In Figure 6, the authors highlighted some parts with a dotted line. Why? Please explain in the Figure legend. Also give a Brief Explanation of the Colors used and what is shown by the different Colors.

Figure 7: The box plots in Figure 7 have a very low Resolution. Some description of the y-axes are not readable, at least for me......

Please enhance the font size. Briefly explain the meaning of the Boxes and bars and describe the subdivision of this Figure.
